# RHGCL: Representation-Driven Hierarchical Graph Contrastive Learning for User-Item Recommendation

## Abstract

Graph Contrastive Learning (GCL), which fuses graph neural networks with contrastive learning, has evolved as a pivotal tool in user-item recommendations. While promising, existing GCL methods often lack explicit modeling of hierarchical item structures, which represent item similarities across varying resolutions. Such hierarchical item structures are ubiquitous in various items (e.g., online products and local businesses), and reflect their inherent organizational properties that serve as critical signals for enhancing recommendation accuracy. In this paper, we propose Representation-driven Hierarchical Graph Contrastive Learning (RHGCL), a novel GCL method that incorporates hierarchical item structures from learned representations for recommendations. First, RHGCL pre-trains a GCL module using cross-layer contrastive learning to obtain user and item representations. Second, RHGCL employs a representation compression and clustering method to construct a two-hierarchy user-item bipartite graph. Ultimately, RHGCL fine-tunes user and item representations by learning on the hierarchical graph, and then provides recommendations based on user-item interaction scores. Experiments on three widely adopted benchmark datasets ranging from 70K to 382K nodes confirm the superior performance of RHGCL over existing baseline models, highlighting the contribution of representation-driven hierarchical item structures in enhancing GCL methods for recommendation tasks.

## 1 Introduction

Graph Contrastive Learning (GCL) represents a key branch of methods in recommendation systems, with notable models such as SGL (Wu et al., 2021a), SimGCL (Yu et al., 2022), and SGRL (He et al., 2024). GCL integrates graph neural networks (GNNs), which capture collaborative signals within user-item graphs (He et al., 2020), with contrastive learning (Chen et al., 2020; Grill et al., 2020), a self-supervised approach that generates representations based on positive and negative sample pairs (Ju et al., 2024; Lin et al., 2022). Since GCL retains the self-supervised representation learning capability, it is advantageous for recommendation systems where supervised signals are sometimes sparse. As examples of GCL, researchers have utilized information bottleneck principles to design graph contrastive modules (Xu et al., 2021a). Besides, user prompts derived from user profiles have been used to enhance GCL-based recommendation frameworks (Yang et al., 2023). Following these methods, various experiments conducted on widely adopted datasets, such as Gowalla (Cai et al., 2023), MovieLens (Lin et al., 2022), and Yelp (Yu et al., 2023a), have demonstrated the superiority of GCL for recommendation tasks.

However, existing GCL-based recommendation methods are still limited. Although certain GCL-based recommendation approaches consider heterogeneous graphs where nodes exhibit various types (Wang et al., 2021b), they ignore hierarchical structures inherent within items in user-item graphs. Here, hierarchical item structures refer to organizations of items with analogous properties at distinct resolutions. For instance, snooker videos and football videos can be categorized under the broader class of sports videos, while sports videos and music audio can be grouped as general entertainment products. Effectively mining these hierarchical dependencies among items can extract valuable item signals for recommendation in two ways. First, the system enables multi-level recommendations, as it could recommend items that are either directly related in the original user-item

graph or exhibit similarity within a broader category (Qin et al., 2023). Second, it effectively mitigates data sparsity issues, because it can recommend items with a few interaction records through the transfer of information from higher-level categories that contain abundant interaction data. Unfortunately, there exists a scarcity of generalizable GCL methods that effectively model hierarchical item structures for user-item recommendation tasks from the following aspects (See Appendix A for more details).

- Recent seminal models GraphLoG (Xu et al., 2021b) and HIGCL (Yan et al., 2023) have incorporated hierarchical item structures for general GCL tasks, rather than focusing on user-item recommendations.
- DiffPool (Ying et al., 2018b) and HiGNN (Li et al., 2020) have highlighted the central role of hierarchical item structures, but not in the context of GCL for recommendations.
- Knowledge graph-based GCL methods, such as HEK-CL (Yuan et al., 2025a) and HKGCL (Liang et al., 2018), reply on knowledge relationships between items to perform recommendation. Despite their effectiveness, a practical barrier to their widespread adoption is the high cost required to construct and maintain up-to-date knowledge graphs.
- Existing representation clustering methods such as NCL (Lin et al., 2022) and HNECL (Wei et al., 2025) apply contrastive learning between node representations and their clustered prototype representations. This tends to reduce embedding uniformity, which is a property shown to be beneficial for prediction performance in graph contrastive learning (Yu et al., 2023a).

In this study, we propose Representation-driven Hierarchical Graph Contrastive Learning (RHGCL), which incorporates hierarchical item structures into GCL for recommendation systems. Specifically, we first obtain user and item representations by pre-training a graph contrastive learning model on user-item graphs, utilizing a weighted combination of prediction loss and contrastive loss. Afterwards, RHGCL compresses item node embeddings and clusters item nodes, thereby constructing user-cluster item graphs. Finally, we derive ultimate user and item embeddings by fine-tuning the model on both the original user-item graphs and the newly user-clustered item graphs. To preserve the uniformity of item node embeddings, we do not include clustered item representations in contrastive learning at this stage. To access the contribution of hierarchical item structures, we conduct experiments comparing RHGCL with state-of-the-art GCL methods, including SGL (Wu et al., 2021a), SimGCL (Yu et al., 2022), and XSimGCL (Yu et al., 2023a), on three publicly available recommendation datasets comprising 70K, 237K, and 382K nodes. The recommendation performance, as indicated by Recall and NDCG scores, demonstrates the superior performance of our RHGCL model.

## 2 RELATED WORK

### 2.1 GRAPH NEURAL NETWORKS FOR RECOMMENDATION

Recommendation systems provide personalized recommendations to web users based on their preferences (Davidson et al., 2010; Kang & McAuley, 2018). Such systems require robust capabilities for learning from complicated user-item interactions (Feng et al., 2025; Lin et al., 2025). These interactions can be seamlessly modeled by GNN methods (Kipf, 2016; Hamilton et al., 2017; Gao et al., 2023), including GCNs (Ying et al., 2018a), NGCF (Wang et al., 2019), and RDL (Robinson et al., 2024). Subsequently, LightGCN simplified NGCF by discarding redundant linear transformation and nonlinear activation functions in NGCF (He et al., 2020). As a pioneering practice, PinSage integrates GNNs with customized random walk-based information fusion for recommendations on Pinterest (Ying et al., 2018a). Recently, OmniSage extended PinSage by combining graph-based relation modeling, sequence-based evolution modeling, and content-based information fusion (Badrinath et al., 2025). In addition to efficient information propagation, node compression emerges as another viable solution in GNNs for recommendation. Notably, LiGNN, developed by LinkedIn, models various entities including members, posts, and jobs as a giant graph (Borisyuk et al., 2024). The used techniques include adaptive multi-hop neighborhood sampling, training data grouping and slicing, and shared-memory queues. Besides, MacGNN aggregates user and item nodes with analogous behavior into macro nodes and learns their representations, supporting online product recommendations on Taobao's homepage (Chen et al., 2024a).

## 2.2 Graph Contrastive Learning for Recommendation

GCL constitutes an unsupervised learning approach that constructs representations from graph data (Veličković et al., 2018). The general process involves creating multiple graph views through graph perturbations (e.g., node deletion, edge modifications, and subgraph sampling), with the strategy of maximizing node representation consistency across these views (You et al., 2020). It facilitates the learning of intrinsic graph structure properties that remain invariant under these perturbations (Zhu et al., 2020). While initial GCL models contain graph perturbation operations, GraphACL discards graph perturbations and leverages one-hop and two-hop neighborhood nodes to develop node representations (Xiao et al., 2023). Recently, GCFormer enhances graph Transformers (Wu et al., 2021b) by incorporating contrastive signals from positive and negative node tokens relative to a target node, exhibiting its superiority over conventional graph Transformers in node classification applications (Chen et al., 2024b).

The self-supervision capability of GCL is well-suited to mine user-item interactions in user-item recommendation systems (Yu et al., 2023b). Specifically, SGL refines supervised user-item graph learning with the contrastive loss (e.g., the InfoNCE loss (Oord et al., 2018)), to improve recommendation accuracy and robustness (Wu et al., 2021a). This seminal work has inspired subsequent GCL methods for recommendation. For example, HCCL integrates local and global collaborative signals within user-item graphs through hypergraphs (Xia et al., 2022). Heterogeneous GCL incorporates heterogeneous relational semantics within graphs (Chen et al., 2023). Another approach, LightGCL, optimizes graph augmentations by applying singular value decomposition to facilitate local and global contrastive learning (Cai et al., 2023). Additionally, industrial researchers have addressed the challenge posed by the long-tail distribution of node degrees by creating pseudo-tail nodes in GCL for recommendation (Zhao et al., 2023).

It is worth mentioning that graph augmentations are not invariably indispensable in GCL for recommendation. Empirical studies have revealed that the uniformity of learned representations plays a central factor in GCL for recommendation (Yu et al., 2022). They proposed SimGCL, incorporating additive uniform noise vectors into node representations to enhance representation uniformity. Furthermore, recommendation researchers refined SimGCL into XSimGCL, which reduces the number of forward and backward information propagation steps from three to one (Yu et al., 2023a). This is achieved by unifying recommendation prediction and contrastive learning together, and contrasting node representations across distinct layers within a single neural network.

## 3 Preliminaries

Following existing studies (Wang et al., 2021a; Yuan et al., 2025b), the user-item recommendation problem is formulated as a link prediction task on a user-item bipartite graph, denoted as $\mathcal{G} = (\mathcal{V}, \mathcal{E})$. Here, the node set $\mathcal{V} = \mathcal{U} \cup \mathcal{I}$ comprises the user set $\mathcal{U}$ and the item set $\mathcal{I}$, where $m = |\mathcal{U}|$ and $n = |\mathcal{I}|$ represent the cardinalities of $\mathcal{U}$ and $\mathcal{I}$, respectively. The edge set $\mathcal{E}$ captures user-item interactions that denote ratings, searches, purchases, and etc. The objective of user-item recommendation is to suggest separate items for each user given observed user-item engagements.

### 3.1 Graph Neural Networks

GNNs serve as critical approaches in user-item recommendation systems by learning representations for $m$ users and $n$ items: $\boldsymbol{E} \in \mathbb{R}^{(m+n) \times d}$, where $d$ denotes the embedding dimension. The matrix $\boldsymbol{E}$ is derived as the final representation matrix within a sequence of representations $\boldsymbol{E}^{(k)}$ where $0 \leq k \leq K$. Here, $K$ signifies the number of layers in the GNN. The information propagation mechanism of a seminal GNN model, LightGCN (He et al., 2020), is expressed as follows:

$$\boldsymbol{E}^{(k+1)} = \boldsymbol{D}^{-\frac{1}{2}} \boldsymbol{A} \boldsymbol{D}^{-\frac{1}{2}} \boldsymbol{E}^{(k)}, \tag{1}$$

where $0 \leq k \leq (K-1)$. $\boldsymbol{A}$ is the adjacency matrix of $\mathcal{G}$. Besides, $\boldsymbol{D}$ denotes the degree matrix of $\mathcal{G}$, where $D_{ll}$ represents the sum of the entries in the $l$-th row of $\boldsymbol{A}$ ($1 \leq l \leq m + n$). To recommend items to a user $i$, we compute the final connecting strength $\hat{y}_{ij}$ between user $i$ and an arbitrary item $j$ as $\hat{y}_{ij} = (\boldsymbol{e}_i)^\top \boldsymbol{e}_j$, where $1 \leq i \leq m, 1 \leq j \leq n$, and $\boldsymbol{e}_i, \boldsymbol{e}_j \in \mathbb{R}^d$ are embeddings extracted from

$\boldsymbol{E}$. We then recommend items to user $i$ by ranking $\hat{y}_{ij}, 1 \leq j \leq n$ in descending order. To train the model, we typically adopt the Bayesian Personalized Ranking loss (Rendle et al., 2012) as the recommendation loss:

$$\mathcal{L}_{rec} = \sum_{(i,j^+,j^-)} -log\left(\sigma(\hat{y}_{i,j^+} - \hat{y}_{i,j^-})\right), \tag{2}$$

where $\sigma(\cdot)$ is the sigmoid function, positive edges $(i, j^+) \in \mathcal{E}$, and negative edges $(i, j^-) \notin \mathcal{E}$. This loss facilitates the updating of learnable node representations (i.e., $\boldsymbol{E}^{(0)}$) to increase or decrease the inner products of endpoint representations associated with positive or negative edges, respectively.

### 3.2 GRAPH CONTRASTIVE LEARNING

The intuition behind GCL for recommendation systems is to learn robust and intrinsic user and item representations that are invariant under graph perturbations where graph structures change (Hassani & Khasahmadi, 2020). This process entails augmenting the original graph $\mathcal{G}$ to generate two views: $\tilde{\mathcal{G}}_1$ and $\tilde{\mathcal{G}}_2$, through operations such as removing nodes or edges given predefined probabilities. The goal of node representation updates is to maximize representation similarity for the same nodes across the two views while minimizing representation similarity of distinct nodes. To achieve this, the InfoNCE loss (Oord et al., 2018) is introduced:

$$\mathcal{L}_{cl} = \sum_{i \in \mathcal{B}} -log\left(\frac{exp(sim(\boldsymbol{e}_{i1}, \boldsymbol{e}_{i2})/\tau)}{\sum_{j \in \mathcal{B}} exp(sim(\boldsymbol{e}_{i1}, \boldsymbol{e}_{j2})/\tau)}\right). \tag{3}$$

Here, $\mathcal{B}$ is a mini-batch. $\boldsymbol{e}_{ik}$ and $\boldsymbol{e}_{jk}$ $(k = 1, 2)$ denote the representations of nodes $i$ and $j$ from $\tilde{\mathcal{G}}_1$ and $\tilde{\mathcal{G}}_2$, respectively. $sim(\cdot, \cdot)$ is a similarity function like the cosine similarity function. $\tau$ refers to temperature parameter. Practically, the GCL module can be fused with GNN module via a hybrid loss $\mathcal{L} = \mathcal{L}_{rec} + \lambda\mathcal{L}_{cl}$ ($\lambda > 0$). Learning guided by loss $\mathcal{L}$ enables the generation of robust self-supervised node representations that align effectively with supervised signals derived from interactions labels. While Eq. 3 represents the complete formulation of GCL, it is noteworthy that some exponents (e.g., positive pairs, and graph augmentations) could be omitted to enhance performance for specific recommendation tasks (Guo et al., 2023; Yu et al., 2022).

## 4 REPRESENTATION-DRIVEN HIERARCHICAL GRAPH CONTRASTIVE LEARNING

In this section, we present RHGCL, which performs graph learning on user-item graphs and user-clustered item graphs. The underlying intuition is to capture hierarchical structures of item nodes while ensuring uniformity of representations.

### 4.1 PRE-TRAINING REPRESENTATIONS ON USER-ITEM BIPARTITE GRAPHS

During the pre-training stage (Fig. 1, left), RHGCL obtains user and item representations using historical user-item interaction records. The objective is to model user and item characteristics without considering hierarchical item structures. To achieve this, we resort to XSimGCL (Yu et al., 2023a), a state-of-the-art GCL approach in user-item recommendation systems.

The model consists of three primary components: (1) graph convolution operator, (2) random noise incorporation, and (3) cross-layer contrastive learning. First, the model passes node representations using light graph convolution operation (i.e., Eq. 1), which captures collaborative filtering signals from user-item interactions. Note that the initial embedding matrix $\boldsymbol{E}^{(0)}$ serves as the collection of learnable representations of users and items. Second, for layer $0 \leq k \leq (K - 1)$, the model introduces a random noise vector $\Delta_i^k$ into the node representations $\boldsymbol{e}_i^k$ on the node $i$:

$$(\boldsymbol{e}_i^k)' = \boldsymbol{e}_i^k + \Delta_i^k, \tag{4}$$

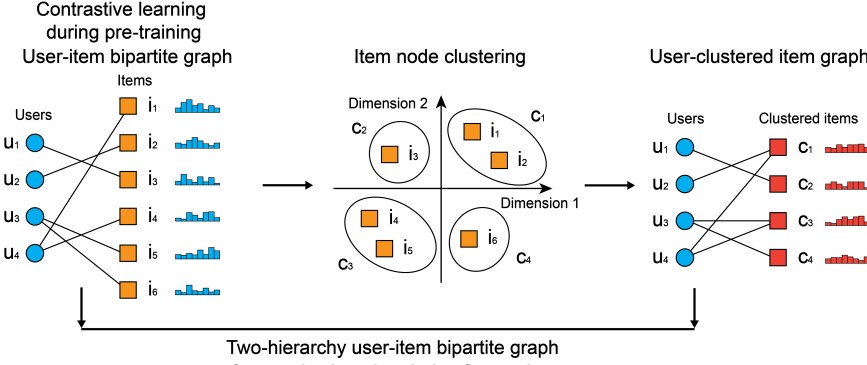

Figure 1: The framework of RHGCL. Initially, RHGCL generates user and item representations using XSimGCL (Yu et al., 2023a), which comprises graph convolution operators with cross-layer contrastive learning (Sec. 4.1). Afterwards, the learned high-dimensional item representations are projected into a two-dimensional latent space using the t-SNE algorithm, forming distinct clusters of item nodes (Sec. 4.2). Building upon these item clusters, RHGCL constructs user-clustered item graphs with clustered items (Sec. 4.3). We then jointly optimize representations of users, items, and clustered items under recommendation and contrastive losses. Finally, RHGCL leverages optimized representations of users, items, and clustered items to infer personalized item rankings for each user.

and propagates $(\boldsymbol{e}_i^k)'$ rather than $\boldsymbol{e}_i^k$ to the next layer of graph convolution. Here, the length of $\Delta_i^k$ is set as a predefined hyperparameter $\epsilon$: $\|\Delta_i^k\| = \epsilon > 0$. The rationale for adding the random noise vector term lies in its capacity to enhance representation uniformity, which improves recommendation performance (Yu et al., 2022). Third, different from Eq. 3, the contrastive learning loss $\mathcal{L}_{cl}$ is constructed by contrasting node representations across different layers within the same graph convolution network.

$$\mathcal{L}_{cl} = \sum_{i \in \mathcal{B}} -log \left( \frac{exp(sim(\boldsymbol{e}_i^K, \boldsymbol{e}_i^{K^*})/\tau)}{\sum_{j \in \mathcal{B}} exp(sim(\boldsymbol{e}_i^K, \boldsymbol{e}_j^{K^*})/\tau)} \right). \tag{5}$$

Here, $0 \leq K^* \leq K$ is a hyperparameter, representing the layer used for contrastive learning with the $K$-th layer. This ingenious design is theoretically supported by spectral graph analysis for contrastive learning (Yu et al., 2023a). Finally, $\mathcal{L}_{cl}$ is integrated with the recommendation loss $\mathcal{L}_{rec}$ to form the hybrid loss $\mathcal{L}$ (Sec. 3.2). Together, after training the model using information passing (Eq. 1) with added noises (Eq. 4), as well as $\mathcal{L}$, we obtain stable pre-trained representations of users as $\boldsymbol{E}_{\mathcal{U}}^{pt} \in \mathbb{R}^{m \times d}$ and items as $\boldsymbol{E}_{\mathcal{I}}^{pt} \in \mathbb{R}^{n \times d}$.

## 4.2 CLUSTERING ITEM NODES

After reaching node representations (i.e., $\boldsymbol{E}_{\mathcal{U}}^{pt}$ and $\boldsymbol{E}_{\mathcal{I}}^{pt}$) from the pre-training stage, the next step of RHGCL is to explicitly model hierarchical item structures. We build these structures by delving into the item similarity (Brovman et al., 2016; Giahi et al., 2023). These item similarity signals play crucial roles in facilitating effective item recommendations for users. For example, a user who has interacted with a basketball video is more likely to prefer a football video over a music audio file. Practically, there are two approaches to evaluate item similarity. First, there exists similarity between two items when they share common attributes (e.g., geolocations, item types). Second, two items are considered similar if they are accessed by similar groups of users and maintain similar node representations (Baltescu et al., 2022). Note that the first approach requires domain-specific attribute information for each user-item recommendation problem, exhibiting weaker generalizability than the second approach. To promote model generalizability, we customize our hierarchical item structures based on the second approach.

**Representation Dimension Reduction.** We first perform the t-distributed Stochastic Neighbor Embedding (t-SNE) algorithm (Van der Maaten & Hinton, 2008) to summarize representations from the

pre-training stage (Fig. 1, middle). The reason for employing the t-SNE algorithm is that it projects high-dimensional vectors to the low-dimensional latent space, while persevering local neighbor relationships[1]. This local neighbor persevering characteristic aligns with representations in recommendation tasks, wherein interacted user and item nodes have similar representations. It is worth noting that Principal Component Analysis is not appropriate here, as it is a linear dimensionality reduction method capturing global data variance. In the t-SNE algorithm (Van der Maaten & Hinton, 2008), for two vectors $\boldsymbol{x}, \boldsymbol{y} \in \mathbb{R}^{d_1}$, the projected vectors $\boldsymbol{x}', \boldsymbol{y}' \in \mathbb{R}^{d_2}$ should satisfy the two conditions: (1) $d_2 < d_1$, (2) $p_{\boldsymbol{xy}}$ is close to $q_{\boldsymbol{x}'\boldsymbol{y}'}$. Here, $p_{\boldsymbol{xy}}$ (or $q_{\boldsymbol{x}'\boldsymbol{y}'}$) quantifies the probability of neighborhood relationship of the two vectors $\boldsymbol{x}, \boldsymbol{y}$ (or $\boldsymbol{x}', \boldsymbol{y}'$) in the original space (or the new space). $p_{\boldsymbol{xy}}$ (or $q_{\boldsymbol{x}'\boldsymbol{y}'}$) is calculated based on the distance of $\boldsymbol{x}$ and $\boldsymbol{y}$ (or $\boldsymbol{x}'$ and $\boldsymbol{y}'$) relative to the neighborhoods using a Student t distribution:

$$p_{\boldsymbol{xy}} = \frac{1/(1 + distance(\boldsymbol{x}, \boldsymbol{y})^2)}{\sum_{\boldsymbol{k},\boldsymbol{l}}(1/(1 + distance(\boldsymbol{k}, \boldsymbol{l})^2))}. \tag{6}$$

Here, $distance(\cdot, \cdot)$ denotes a distance metric, such as the squared Euclidean distance. $\boldsymbol{k}, \boldsymbol{l} \in \mathbb{R}^{d_1}$ are within the neighborhood of $\boldsymbol{x}, \boldsymbol{y}$. The number of selected neighborhood is measured by a model hyperparameter: perplexity. By optimizing $\boldsymbol{x}', \boldsymbol{y}'$ in the two-dimensional space, we are able to convert item node representations from $\boldsymbol{E}_{\mathcal{I}}^{pt} \in \mathbb{R}^{n \times d}$ to $\boldsymbol{E}_{\mathcal{I}}^{tsne} \in \mathbb{R}^{n \times 2}$. Note that the time complexity of the t-SNE algorithm is $\mathcal{O}(n^2)$. This can be optimized to $\mathcal{O}(n \log(n))$ from the Barnes-Hut SNE algorithm (Van Der Maaten, 2013) for large graphs.

**Clustering Item Nodes in the Two-Dimensional Space.** Based on $\boldsymbol{E}_{\mathcal{I}}^{tsne}$, we now cluster item nodes in the two-dimensional space. The goal is to form simple and interpretable item node clusters for hierarchical item structures. Given the connection difference between distinct user-item graphs, we introduce task-specific hyperparameter $\rho \geq 1$ and $\theta \geq 1$ to perform item node clustering. Here, $\rho$ and $\theta$ represent the number of divisions along the radial and angular directions, respectively. The item nodes whose two-dimensional representations are located within the same sector are clustered into a group. This setting leads to the total number of clusters as $\rho * \theta$. For example, in Fig. 1, middle panel, there are one radial division ($\rho = 1$) and four angular divisions ($\theta = 4$), leading to total $1 * 4 = 4$ clusters of item nodes. Different from the seminal hierarchical GNN study DiffPool (Ying et al., 2018a) with soft clustering, our approach utilizes deterministic clustering for item nodes to ensure simplicity and enhance interpretability.

### 4.3 FINE-TUNING REPRESENTATIONS ON TWO-HIERARCHY USER-ITEM BIPARTITE GRAPHS

After representation compressions and item node clustering, we now build user-clustered item graphs (Fig. 1, right). We create the clustered item set $\mathcal{C}$, which contains the derived clusters from item node clustering. Hence, $|\mathcal{C}| = \rho * \theta$. In this way, we have extended our original node set $\mathcal{V} = \mathcal{U} \cup \mathcal{I}$ to $\mathcal{V}^c = \mathcal{U} \cup \mathcal{C}$. We now delineate edges between nodes in $\mathcal{U}$ and $\mathcal{C}$. Particularly, the user node $u \in \mathcal{U}$ is linked to the clustered item node $c \in \mathcal{C}$ if and only if the user node $u$ is connected to at least one item node within the cluster represented by $c$. For instance, node $u_2$ is connected to $c_1$ in the user-clustered item graph (Fig. 1, right), because $u_2$ is linked to $i_2$ in the user-item graph (Fig. 1, left), and $i_2$ is a member of cluster $c_1$ (Fig. 1, middle).

Until now, we have obtained two graphs: the user-item graph (i.e., $\mathcal{V}$), and the user-clustered item graph (i.e., $\mathcal{V}^c$), which are collectively called the two-hierarchy user-item bipartite graph. We perform graph convolution operators (i.e., Eq. 1) on the two graphs separately, and finally obtain three fine-tuned representation matrices: $\boldsymbol{E}_{\mathcal{U}}^{ft} \in \mathbb{R}^{m \times d}$, $\boldsymbol{E}_{\mathcal{I}}^{ft} \in \mathbb{R}^{n \times d}$, and $\boldsymbol{E}_{\mathcal{C}}^{ft} \in \mathbb{R}^{(\rho * \theta) \times d}$. The final connecting strength $y_{ij}$ between user $i$ and item $j$ is defined as:

$$\hat{y}_{ij} = (\boldsymbol{e}_{\mathcal{U},i}^{ft})^{\top}(\boldsymbol{e}_{\mathcal{I},j}^{ft} + \sum_{1 \leq k \leq \rho * \theta} w_{jk}\boldsymbol{e}_{\mathcal{C},k}^{ft}). \tag{7}$$

---

[1]https://distill.pub/2016/misread-tsne/

Here, $e_{\mathcal{U},i}^{ft} \in \mathbb{R}^d$ denotes the $i$-th row of $E_{\mathcal{U}}^{ft}$, which corresponds to the user representation. Similarly, $e_{\mathcal{I},j}^{ft} \in \mathbb{R}^d$ represents the $j$-th row of $E_{\mathcal{I}}^{ft}$, representing the item representation, and $e_{\mathcal{C},k}^{ft} \in \mathbb{R}^d$ denotes the $k$-th row of $E_{\mathcal{C}}^{ft}$, indicating the clustered item representation. The term $w_{jk}$ denotes the clustering membership, where $w_{jk} = 1$ if the item node $j$ belongs to cluster $k$, and $w_{jk} = 0$ otherwise. To train the model, we set the hybrid loss:

$$\mathcal{L} = \mathcal{L}_{rec}^{ui} + \mathcal{L}_{rec}^{uc} + \lambda \mathcal{L}_{cl}^{ui}. \tag{8}$$

Here, $\mathcal{L}_{rec}^{ui}$ and $\mathcal{L}_{rec}^{uc}$ represent the recommendation loss on the user-item graph and the user-clustered item graph, respectively, while $\mathcal{L}_{cl}^{ui}$ is the cross-layer contrastive learning loss on the user-item graph. Notably, we do not include the contrastive learning term between item representations and clustered item representations (i.e., $\mathcal{L}_{cl}^{ic}$), as it may compromise embedding uniformity. Furthermore, we provide that the time complexity of RHGCL is slightly higher than that of XSimGCL, due to the incorporation of edges in the user-clustered item graph $\mathcal{V}^c$ (Appendix E).

## 5 EXPERIMENTS

To evaluate the performance of RHGCL, we conduct a series of experiments utilizing three publicly available datasets. The target of our experiments is to answer the following questions:

- **Q1.** What is the performance of RHGCL in various user-item recommendation tasks?
- **Q2.** Do our hierarchical item structures enhance model performance?
- **Q3.** How does the number of cluster divisions (i.e., $\rho$ and $\theta$) affect prediction accuracy?

### 5.1 EXPERIMENT SETTINGS

#### 5.1.1 DATASETS AND BASELINE MODELS

We employ three classical user-item datasets[2]: **Yelp2018**, **Amazon-Kindle**, and **Alibaba-iFashion** to perform recommendation tasks. These datasets haven been extensively used in previous user-item recommendation studies (Wang et al., 2019; Yu et al., 2023a; 2022) (See Appendix B for statistics). Given that existing model comparisons (He et al., 2020; Yu et al., 2023a) have demonstrated that the models Mult-VAE (Liang et al., 2018), NGCF (Wang et al., 2019), and MixGCF (Huang et al., 2021) exhibit relatively inferior performance compared to state-of-the-art methods, we select the following eight models as baselines: **BUIR** (Lee et al., 2021), **DNN+SSL** (Yao et al., 2021), **Light-GCN** (He et al., 2020), **NCL** (Lin et al., 2022), **SGL** (Wu et al., 2021a), **SimGCL** (Yu et al., 2022), **XSimGCL** (Yu et al., 2023a), and **HNECL** (Wei et al., 2025) (See Appendix C for details).

#### 5.1.2 PARAMETERS

For BUIR and DDN+SSL, we resort to implementation outcomes in the existing study (Yu et al., 2023a). The hyperparameters for LightGCN, SGL, NCL, SimGCL, XSimGCL, and HNECL include the hidden dimension $d$, the number of layers $K$, the temperature parameter $\tau$, and the learning weight associated with the contrastive learning loss $\lambda$. To ensure a fair comparison, the hyperparameters for the Yelp2018 dataset were configured in accordance with the SELFRec repository[3]. The hyperparameters for the Amazon-Kindle and Alibaba-iFashion datasets were set to align with those used in the study XSimGCL (Yu et al., 2023a). For HNECL, we set the configuration of hierarchical prototypes as (2000, 1000, 100). Since our RHGCL method is built upon XSimGCL, we keep consistent hyperparameters betwen XSimGCL and RHGCL (Table 1). Finally, the embedding dimension $d$ is set to 64, the batch size to 2048, and the learning rate to 0.0001. To train the model, we use the Xavier initialization method (Glorot & Bengio, 2010) and refer to the Adam optimizer (Kingma & Ba, 2014). The maximum number of training epochs is set to 50 for Yelp2018, 100 for Amazon-Kindle, and 50 for Alibaba-iFashion, respectively. The training and testing time of RHGCL on a single CPU is around 20, 40, and 60 hours for Yelp2018, Amazon-Kindle, and Alibaba-iFashion, respectively.

---

[2]https://github.com/Coder-Yu/SELFRec/tree/main/dataset/
[3]https://github.com/Coder-Yu/SELFRec

Table 1: Hyperparameters in RHGCL. $K$: the number of layers; $K^\star$: the layer selected for contrastive learning; $\lambda$: the weight for contrastive loss; $\epsilon$: the length of added noise vectors; $\tau$: temperature parameter; $\rho$: the number of representation divisions along the radial direction; $\theta$: the number of representation divisions along the angular direction.

| Dataset | $K$ | $K^\star$ | $\lambda$ | $\epsilon$ | $\tau$ | $\rho$ | $\theta$ |
|---|---|---|---|---|---|---|---|
| Yelp2018 | 3 | 1 | 0.20 | 0.20 | 0.15 | 8 | 2 |
| Amazon-Kindle | 3 | 1 | 0.20 | 0.10 | 0.20 | 1 | 2 |
| Alibaba-iFashion | 4 | 4 | 0.05 | 0.05 | 0.20 | 4 | 8 |

## 5.2 OVERALL PERFORMANCE

Table 2: Performance comparison. Note that the performance metrics of BUIR and DNN+SSL are sourced from the existing study (Yu et al., 2023a). For the remaining four baseline methods, we conduct the experiments using our own computational resources. We use underlining to denote the second-best performance. The reported improvement is calculated based on RHGCL and the second-best outcome.

| | Yelp2018 | | Amazon-Kindle | | Alibaba-iFashion | |
|---|---|---|---|---|---|---|
| | Recall@20 | NDCG@20 | Recall@20 | NDCG@20 | Recall@20 | NDCG@20 |
| BUIR (Lee et al., 2021) | 0.0487 | 0.0404 | 0.0922 | 0.0528 | 0.0830 | 0.0384 |
| DNN+SSL (Yao et al., 2021) | 0.0483 | 0.0382 | 0.1520 | 0.0989 | 0.0818 | 0.0375 |
| LightGCN (He et al., 2020) | 0.0595 | 0.0487 | 0.1900 | 0.1208 | 0.0885 | 0.0410 |
| SGL (Wu et al., 2021a) | 0.0677 | 0.0555 | 0.2098 | 0.1324 | 0.1138 | 0.0541 |
| NCL (Lin et al., 2022) | 0.0674 | 0.0553 | 0.1857 | 0.1147 | 0.0897 | 0.0417 |
| SimGCL (Yu et al., 2022) | 0.0727 | 0.0598 | 0.2031 | 0.1264 | 0.1176 | 0.0561 |
| XSimGCL (Yu et al., 2023a) | 0.0726 | 0.0597 | 0.2061 | 0.1324 | 0.1189 | 0.0569 |
| HNECL (Wei et al., 2025) | 0.0674 | 0.0554 | 0.1823 | 0.1096 | 0.0976 | 0.0454 |
| RHGCL | **0.0736** | **0.0605** | **0.2117** | **0.1354** | **0.1194** | **0.0572** |
| Improvement | +1.2% | +1.2% | +2.7% | +2.3% | +0.4% | +0.5% |

To examine the recommendation performance of RHGCL, we present recommendation metrics, Recall@20 and NDCG@20 scores on the testing sets of the three datasets in Table 2. Among the nine models, BUIR and DNN+SSL yield the lowest Recall@20 and NDCG@20 scores across all three datasets. This is because these models have limited capability to capture multi-hop user-item similarity within historical interactions. In contrast, LightGCN brings improved prediction performance, as this model retains essential graph convolution operations while eliminating redundant linear transformations and nonlinear activations in recommendation tasks. As one of the pioneering self-supervised models in graph learning, SGL significantly enhances prediction accuracy. This improvement can be attributed to the InfoNCE loss function, which pulls node representations associated with positive samples while pushing those linked to negative samples. NCL has considered both the structural and semantic neighbors in contrastive learning, leading to competitive prediction performance. Compared to NCL, HNECL yields similar model performance. This observation aligns with the findings reported in the HNECL paper which shows that increasing the number of hierarchical layers provides little additional improvement (Wei et al., 2025). Furthermore, SimGCL and XSimGCL exhibit the overall highest Recall@20 and NDCG@20 scores among baseline models. This achievement is supported by their ability to promote representation uniformity through added noise vectors.

Notably, our proposed RHGCL model attains the highest Recall@20 and NDCG@20 scores among the nine models, highlighting its superior capability in modeling user-item relationships in recommendation tasks (Answer **Q1**). Recall that RHGCL is an extension of XSimGCL, incorporating hierarchical item structures. Hence, the performance comparison between RHGCL and XSimGCL further substantiates the beneficial contribution of hierarchical item structures in user-item recommendation (Answer **Q2**). This conclusion is further confirmed in Appendix D, where we reveal that positive user-item pairs exhibit greater distinguishability from negative user-item pairs after fine-tuning compared to the distinguishability observed after pre-training.

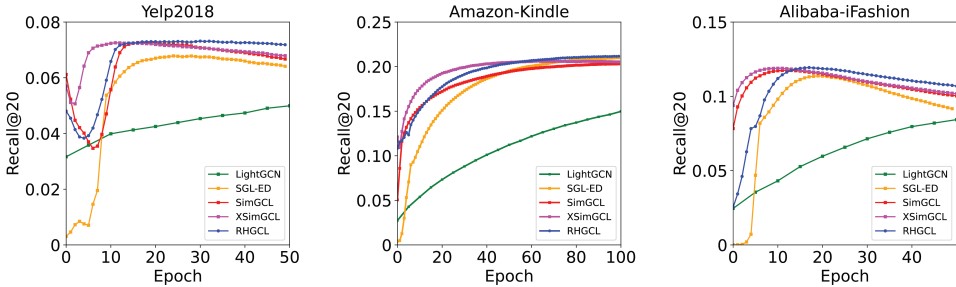

Figure 2: Learning curves of different models on the three datasets.

To analyze the training dynamics of different models, we visualize the evolution of Recall@20 scores in Fig. 2. We see that the learning curves of RHGCL exhibit a trend similar to that of SimGCL and XSimGCL. On the Yelp2018 dataset, the Recall@20 score of RHGCL first declines before subsequently increasing. For the Amazon-Kindle dataset, the score of RHGCL exhibits steady upward trend. Finally, there is an increasing-then-decreasing pattern on the Alibaba-iFashion dataset. Notably, RHGCL consistently achieves highest prediction accuracy. This can be reflected by its ability to (1) reach larger Recall@20 scores; and (2) sustain larger scores over a longer span of training epochs, particularly on the Yelp2018 dataset.

## 5.3 SENSITIVITY STUDY

In this subsection, we investigate the influence of the spatial division parameters $\rho$ and $\theta$ on prediction performance in Fig. 3. Recall that $\rho$ and $\theta$ denote the number of radial and angular divisions, respectively. The results show that setting $\theta = 2$ on the Yelp2018 and Amazon-Kindle datasets yield optimal performance, whereas $\theta = 8, \rho = 4$ leads to highest accuracy on the Alibaba-iFashion dataset. These findings suggest that the best hyperparameter settings are task-specific, depending on user-item interaction properties (Answer **Q3**). On one hand, increasing $\rho$ and $\theta$ results in a greater number of divisions and item clusters, enabling finer-grained item cluster modeling. On the other hand, a larger number of divisions may lead to more boundary-adjacent items and cut off item interactions across division sector boundaries.

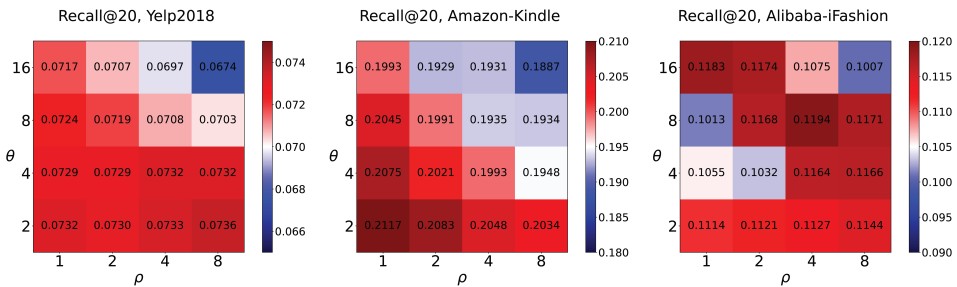

Figure 3: Recall@20 scores from RHGCL with different values of hyperparameters $\rho$ and $\theta$.

We also investigate the influence of perplexity, which measures the number of selected neighborhoods in the t-SNE algorithm, on recommendation performance. A higher value of perplexity implies that the algorithm considers a wider range of neighborhoods when projecting representations. As illustrated in Fig. 4, the perplexity setting of 30 yields the highest Recall@20 and NDCG@20 scores among the five tested values on the Yelp2018 and Alibaba-iFashion datasets. In contrast, the Recall@20 and NDCG@20 scores on the Amazon-Kindle dataset exhibit only moderate sensitivity to changes in perplexity. These findings suggest that tuning the perplexity parameter can be a viable strategy for enhancing recommendation when applying RHGCL to other datasets.

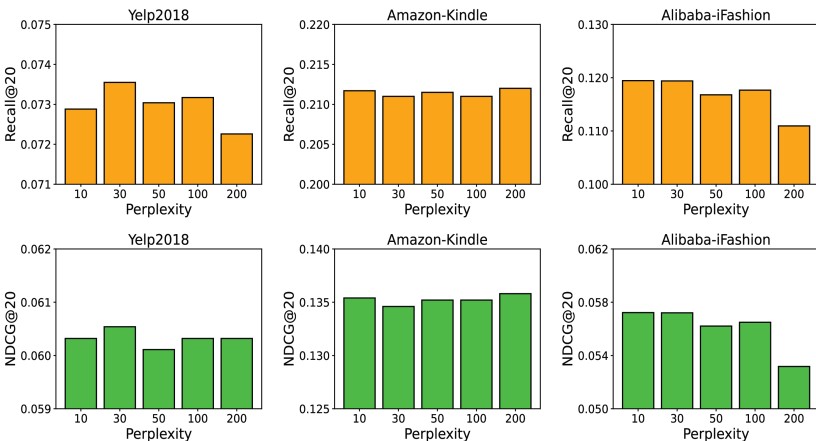

Figure 4: Recall@20 and NDCG@20 scores from RHGCL with different values of perplexity in t-SNE.

## 6 CONCLUSION

This study proposes a generalizable graph contrastive learning approach, called RHGCL, for user-item recommendation tasks. RHGCL performs contrastive learning on node representations during both the pre-training phase, utilizing the original user-item graph, and the fine-tuning phase, employing a two-hierarchy user-item bipartite graph. Note that our method functions as an extension of existing GCL methods, thereby ensuring high generalizability. Experimental results on three open user-item datasets confirm the positive contribution of hierarchical item structures and overall superior performance of RHGCL. One limitation of this study is that the superiority of our proposed RHGCL method has not been demonstrated on large-scale industrial recommendation datasets that contain over millions of user and item nodes. Future studies can explore the role of hierarchical item structures in such scenarios. Moreover, the semantic interpretability of the clustered items is limited, due to the scarcity of item category information in the three datasets. This gap can be mitigated when datasets with richer categorical annotations are available.

ACKNOWLEDGMENTS

This study is supported by our anonymous agencies.

THE USE OF LARGE LANGUAGE MODELS

We have employed Large Language Models to polish texts in this paper.

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

## A COMPARISON WITH EXISTING STUDIES

Table 3: Comparison between our model and recent studies on hierarchical graph contrastive learning for recommendation systems. U-I: User-Item. KGs: Knowledge Graphs. U-U: User-User. I-I: Item-Item.

| Study | Target | Graph learning | Contrastive learning | Hierarchy |
|---|---|---|---|---|
| HKGCL (Li et al., 2023) | U-I ratings | Along KGs | Cross domains | From domains |
| HGCL (Shui et al., 2024) | U-I ratings | Along subgraphs | Cross subgraphs | From node neighbors |
| ReHCL (Wang et al., 2024) | U-I ratings | Along topic&semantic graphs | Cross-views&cross-modal | From review text data |
| HEK-CL (Yuan et al., 2025a) | U-I interactions | Along KGs | Cross graphs and KGs | From KGs |
| HNECL (Wei et al., 2025) | U-I interactions | Along U-I, U-U, I-I graphs | Cross layers | From two-hop neighbors |
| RHGCL | U-I interactions | Along U-I graphs | Cross layers | From entire graphs |

We summarize recent hierarchical GCL approaches for recommendation systems in Table 3 to clarify the distinctions between our proposed RHGCL models and recent methods. In particular, recent models can be categorized into three types.

- **Incorporating KGs.** HKGCL (Li et al., 2023) conducts graph learning based on edges within KGs, whereas HEK-CL (Yuan et al., 2025a) develops its contrastive learning module by leveraging the structural properties of KGs. The two models are effective when the external item-relationship data is available to construct KGs. In contrast, our approach relies on only U-I interactions, which are generally more accessible.

- **Utilizing review data.** ReHCL (Wang et al., 2024) employs review data—which contains rich semantic information from users regarding items—to conduct hierarchical GCL on U-I ratings. While useful, the availability of user review data is sometimes limited, often due to issues such as privacy considerations.

- **Relying on node neighbors.** The models HGCL (Shui et al., 2024) and HNECL (Wei et al., 2025) establish hierarchical structures for contrastive learning by exploiting node neighborhood relationships. Different from the two models, our proposed RHGCL model derives hierarchical structures by clustering node representations across the entire set of items, thereby enabling a more comprehensive capture of global item-level information.

## B DATASETS

We use the following three datasets to examine the performance of RHGCL.

- **Yelp2018. training set:** 31,668 users; 38,048 items (i.e., local business); 1,237,259 user-item interactions. **testing set:** 31,668 users; 36,073 items; 324,147 user-item interactions.

- **Amazon-Kindle. training set:** 138,333 users; 98,572 items (i.e., books); 1,527,913 interactions. **testing set:** 107,593 users; 82,935 items; 382,052 interactions.

- **Alibaba-iFashion. training set:** 300,000 users; 81,614 items (i.e., garments); 1,255,447 interactions; **testing set:** 268,847 users; 48,483 items; 352,366 interactions.

## C BASELINE MODELS

The eight baseline models are listed below.

- **BUIR** is a two-encoder representation framework without negative sampling (Lee et al., 2021).

- **DNN+SSL** employs data augmentation methods to handle data sparsity problems (Yao et al., 2021).

- **LightGCN** represents a streamlined version of GNNs, removing linear transformation and nonlinear activation operations (He et al., 2020).

- **NCL** incorporates the potential neighbors of a node from structural and semantic spaces into contrastive learning (Lin et al., 2022).

- **SGL** introduces self-supervision into graph learning by leveraging multiple views derived from the original user-item graphs (Wu et al., 2021a).

- **SimGCL** enhances SGL by adding random noises during the model training process (Yu et al., 2022).

- **XSimGCL** simplifies SimGCL by unifying graph-based recommendation and contrastive learning into a single information propagation (Yu et al., 2023a).

- **HNECL** clusters node embeddings into hierarchical prototypes and conducts contrastive learning between user and item embeddings across distinct graph convolution layers (Wei et al., 2025).

## D    CONNECTING STRENGTH ANALYSIS

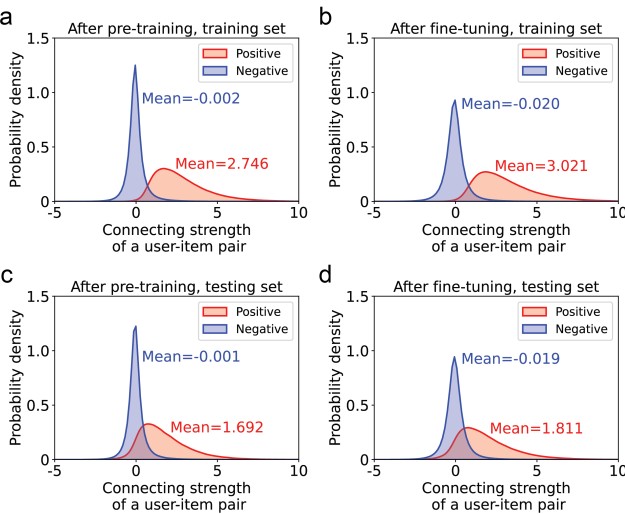

Figure 5: The connecting strength of positive and negative user-item pairs after pre-training ($a$, $c$) and fine-tuning ($b$, $d$) on the Yelp2018 dataset.

To further validate the contribution of the item clustering and fine-tuning modules in RHGCL, we track the changes in the connection strength of user-item pairs. As described in Sec. 3.1, the connecting strength (i.e., $\hat{y}_{ij}$) is defined as the dot product between the representations of user $i$ and item $j$, where a higher value of $\hat{y}_{ij}$ corresponds to an increased probability of recommendation.

We visualize the probability density distribution of the connecting strength for positive and negative user-item pairs on the Yelp2018 dataset in Fig. 5. Here, positive pairs refer to user-item pairs observed in the training or testing sets. Negative pairs are generated by randomly sampling ten items from the entire item set for each user in the training or testing sets. Note that Fig. 5$a$ and $b$ correspond to the training set, whereas Fig. 5$c$ and $d$ correspond to the testing set. Comparing Fig. 5$a$ and $b$, we can see that the average connecting strength increases from 2.746 to 3.021 for the positive user-item pairs, and decreases from -0.002 to -0.020 for the negative user-item pairs. These changes indicate that positive pairs are more distinguishable with negative pairs after fine-tuning than after pre-training. Similarly, on the testing set (Fig. 5$c$ and $d$), we observe an increase of average connecting strength for positive edges (from 1.692 to 1.811) and a decrease of average connecting strength for negative edges (from -0.001 to -0.019). All these results provide strong empirical evidence that our designed item clustering and fine-tuning modules in RHGCL effectively enhance the recommendation performance.

# E  TIME COMPLEXITY ANALYSIS

In this section, we present the time complexity analysis of our proposed RHGCL and contrast it with XSimGCL. To clarify, we first summarize key notations. Recall from Sec. 3 that $|\mathcal{E}|$ represents the number of edges in the user-item bipartite graph. $K$ is the number of layers in graph convolution operations and $d$ represents the node embedding dimension. We introduce $|\mathcal{E}^{uc}|$ to denote the number of edges between the user set $\mathcal{U}$ and the clustered item set $\mathcal{C}$. Hence, we have:

$$|\mathcal{E}^{uc}| \leq |\mathcal{E}|, |\mathcal{E}^{uc}| \leq m\rho\theta. \tag{9}$$

The first inequality holds since each user-item pair in $\mathcal{E}$ is mapped to a user-clustered item pair in $\mathcal{E}^{uc}$. The second inequality follows from the fact that the number of edges in $\mathcal{U} \cup \mathcal{C}$ is upper bounded by the size of $\mathcal{U}$ (i.e., $m$) and the size of $\mathcal{C}$ (i.e., $\rho\theta$). Further, we use $b$ to represent the batch size and $B$ to denote the number of nodes within a batch.

Table 4: Time complexity comparison between XSimGCL and RHGCL.

| Model | Graph convolution | Recommendation loss | Contrastive loss |
|---|---|---|---|
| XSimGCL | $\mathcal{O}(2|\mathcal{E}|Kd)$ | $\mathcal{O}(2bd)$ | $\mathcal{O}(bBd)$ |
| RHGCL | $\mathcal{O}(2|\mathcal{E}|Kd) + \mathcal{O}(2|\mathcal{E}^{uc}|Kd)$ | $\mathcal{O}(2bd)$ | $\mathcal{O}(bBd)$ |

Accordingly, we summarize the time complexity of RHGCL and XSimGCL in Table 4. Recall from Sec. 4.3 that we perform graph convolution operations on the user-item graph and the user-clustered item graph separately. This introduces an additional graph convolution time complexity $\mathcal{O}(2|\mathcal{E}^{uc}|Kd)$, which is theoretically upper bounded by $\mathcal{O}(2|\mathcal{E}|Kd)$ and is empirically small given the small number of clusters (i.e., $\rho\theta$). Moreover, the time complexity of the recommendation loss calculation in RHGCL scales linearly with both $b$ and $d$, which is the same as those from XSimGCL. Recall from Sec. 4.3 that the contrastive loss is computed over $\mathcal{U}$ and $\mathcal{I}$, leading to the same time complexity $\mathcal{O}(bBd)$ as XSimGCL. Together, RHGCL introduces only a minor increase in time complexity over XSimGCL.

