# OpenReview forum: "RHGCL: Representation-Driven Hierarchical Graph Contrastive Learning for User-Item Recommendation"
_ICLR.cc/2026/Conference — Submitted to ICLR 2026_

### Official Review · Reviewer_myzr · 2025-10-31

**Soundness:** 2
**Presentation:** 1
**Contribution:** 2
**Rating:** 2
**Confidence:** 4

**Summary:**

This paper proposes RHGCL, a hierarchical graph contrastive learning method for user-item recommendation that incorporates hierarchical item structures. The approach consists of three stages: (1) pre-training using XSimGCL to obtain user and item representations, (2) clustering items in a 2D space using t-SNE and geometric partitioning, and (3) fine-tuning on both the original user-item graph and a user-clustered item graph. Experiments on three datasets show improvements over baseline methods.

**Strengths:**

1. The paper identifies a relevant gap in existing GCL methods regarding hierarchical item structures and provides concrete examples of why this matters for recommendation systems.

2. The method demonstrates performance gains across all three datasets and metrics, suggesting the approach has merit.

3. The paper includes appropriate baselines, multiple datasets of varying sizes, and sensitivity analyses for key hyperparameters.

**Weaknesses:**

1. The core contribution appears to be applying t-SNE dimensionality reduction followed by geometric clustering (radial/angular divisions) to create hierarchical structures. This is a relatively straightforward extension of XSimGCL without significant methodological innovation.

2. The paper does not provide theoretical or empirical justification for why t-SNE is optimal for capturing hierarchical item relationships over alternatives like UMAP, spectral embedding, or learned hierarchical representations.

3. The deterministic polar coordinate partitioning seems arbitrary and overly simplistic. Why should semantic item relationships align with geometric sectors in a 2D embedding space? This lacks theoretical grounding.

4. No comparison with methods that learn hierarchical structures (e.g., differentiable pooling variations adapted for bipartite graphs).

5. The related work section (Appendix A) lists several hierarchical GCL methods for recommendations (HKGCL, HGCL, ReHCL, HEK-CL, HNECL), but none are included in the experimental comparison. While the authors claim these methods require knowledge graphs or review data, this justification is insufficient: At minimum, methods that only require interaction data (like HNECL) should be compared.


6. The improvements over XSimGCL are very modest.

**Questions:**

1. Can you provide theoretical or empirical justification for why t-SNE followed by geometric clustering is superior to alternative approaches for capturing hierarchical item structures?

2. Why not compare with other hierarchical recommendation methods, particularly those that don't require external knowledge graphs (e.g., HNECL)?

3. Can you provide statistical significance tests demonstrating that the observed improvements are not due to random variation?

4. How do the learned clusters correspond to actual item categories or semantic relationships? Can you provide qualitative analysis?

---

> ### Author Response · Authors · 2025-12-01
> **Official Reply to Reviewer myzr**
>
> **Reply to Weakness 1.** We sincerely thank Reviewer myzr for the question regarding methodological innovations.
>
> RHGCL comprises three components: pre-training (Sec. 4.1), item node clustering (Sec. 4.2), and fine-tuning (Sec. 4.3). We acknowledge that the pre-training stage is based on XSimGCL. However, the subsequent two modules, item node clustering and fine-tuning, are novel. Together, they establish a new paradigm in which fine-tuning with clustered embedding structures enhances recommendation performance on the original user-item graph.
>
> **Reply to Weakness 2 and Question 1.** We appreciate that Reviewer myzr raised the concern about alternatives like UMAP (Uniform Manifold Approximation and Projection), spectral embedding, and learned hierarchical representations. We argue that t-SNE followed by geometric clustering is more suitable for user-item recommendation problems.
>
> t-SNE preserves local structures of original high-dimensional embeddings via its perplexity hyperparameter. This aligns with the user-item recommendation task where local items are recommended to the ego user node.
>
> However, UMAP and spectral embeddings prioritize global information, which is reflected by their eigenvector computation of graph Laplacian: $D^{-1/2}(D-A)D^{-1/2}$. Here $D$ and $A$ represent degree and adjacency matrices. Furthermore, we tested alternative baselines, including NCL (https://arxiv.org/pdf/2202.06200; WWW 2022) and HNECL (Knowledge-Based Systems, 2025) with learned hierarchical representations through K-Means. The results show that their performance does not beat RHGCL (Reply to Weaknesses 1 and 2 to Reviewer p9ZG).
>
> **Reply to Weakness 3 and Question 4.** We sincerely thank Reviewer myzr for the questions regarding the semantic item relationships and item categories.
>
> Deriving semantic item relationships and item categories from clustered item representations is a common challenge for existing hierarchical models like NCL and HNECL. These studies tested the number of clusters $K$ up to the thousands but provided little interpretation of the semantic meanings of items.
>
> We acknowledge that our interpretations of semantic item meanings are also limited. In Conclusion, we added the following texts to acknowledge this.
>
> *Moreover, the semantic interpretability of the clustered items is limited, due to the scarcity of item category information in the three datasets. This gap can be mitigated when datasets with richer categorical annotations are available*.
>
> **Reply to Weakness 4.** We are grateful to Reviewer myzr for suggesting that we compare RHGCL with methods that explicitly learn hierarchical structures. In response, we examined two representative approaches: NCL and HNECL .
>
> NCL employs the K-Means algorithm to derive clustered prototype representations (Equations 10 and 11 in the NCL paper). Similarly, HNECL utilizes the hierarchical K-Means algorithm to obtain clustered representations across multiple hierarchical layers (Section 3.4.1 in the HNECL paper). In both methods, the clustered prototype representations are updated dynamically during model training.
>
> As detailed in our Reply to Weaknesses 1 and 2 to Reviewer p9ZG, experimental results demonstrate that both NCL and HNECL underperform RHGCL. This is because NCL and HNECL perform contrastive learning between node representations and their clustered prototype representations, which reduces embedding uniformity. Here, embedding uniformity is a property known to enhance prediction performance (https://arxiv.org/pdf/2209.02544).
>
> **Reply to Weakness 5 and Question 2.** We sincerely thank Reviewer myzr for suggesting a comparison between RHGCL and HNECL, which is built on learned representations without knowledge graphs.
>
> To address this, we implemented HNECL. The prediction results indicate that HNECL achieves lower performance than RHGCL (Reply to Weaknesses 2 to Reviewer p9ZG).
>
> **Reply to Weakness 6 and Question 3.** We appreciate Reviewer myzr for raising concerning the extent of RHGCL's improvement over XSimGCL. To address this, we added the relative improvement of RHGCL over best{SimGCL, XSimGCL} in Table 2: 1.2%, 1.2%, 2.7%, 2.3%, 0.4%, and 0.5%. We also conducted one-sided t-test. We find the improvement of RHGCL over best{SimGCL, XSimGCL} is statistically significant on Amazon-Kindle with p-values < 0.01 and not statistically significant on Yelp2018 and Alibaba-iFashion.
>
> |--------------------| ------------------Yelp2018|---------Amazon-Kindle|--------Alibaba-iFashion|
> |-----------Model|Recall@20|NDCG@20|Recall@20|NDCG@20|Recall@20|NDCG@20|
> |---------SimGCL|-----0.0727|-----0.0598|-----0.2031|------0.1264|-----0.1176|------0.0561|
> |-------XSimGCL|-----0.0726|-----0.0597|-----0.2061|------0.1324|-----0.1189|------0.0569|
> |----------RHGCL|-----0.0736|-----0.0605|-----0.2117|------0.1354|-----0.1194|------0.0572|
> |Improvement|--------1.2%|--------1.2%|-------2.7%|---------2.3%|--------0.4%|---------0.5%|

---

### Official Review · Reviewer_4gjn · 2025-11-01

**Soundness:** 3
**Presentation:** 3
**Contribution:** 3
**Rating:** 8
**Confidence:** 3

**Summary:**

The paper presents a novel Graph Contrastive Learning (GCL) method for user-item recommendation that incorporates hierarchical item structures. It pre-trains a GCL module, constructs a two-hierarchy user-item bipartite graph via representation compression and clustering, and then fine-tunes representations. Experiments show RHGCL outperforms existing models by enhancing GCL with representation-driven hierarchical item structures for recommendation tasks

**Strengths:**

1. an effective method with neat writing.
2. The model consistently outperforms strong baselines (e.g., XSimGCL, SimGCL, LightGCN) across three benchmark datasets (Yelp2018, Amazon-Kindle, Alibaba-iFashion), with meaningful improvements in Recall@20 and NDCG@20
3. The paper presents a complete pipeline—pretraining, hierarchical clustering (via t-SNE), and fine-tuning on dual-level graphs—illustrating conceptual clarity and reproducibility.

**Weaknesses:**

1. As mentioned by the authors, incorporating hierarchical information into a recommendation system is not new. The novelty is my main concern.
2. Adding hierarchical information will incur more computation; it is better to provide some analysis.
3.  Using t-SNE followed by radial–angular partitioning introduces hyperparameters  that may not generalise well. The clustering might be unstable or computationally expensive for large graphs.

**Questions:**

The questions are related to the weakness.
1. what's the core novelty compares to other works except combining different small pieces?
2. computational anlaysis?
3. ablation study on clustering?

---

> ### Author Response · Authors · 2025-12-01
> **Official Reply to Reviewer 4gjn**
>
> **Reply to Weakness 1 and Question 1.**  We appreciate Reviewer 4gjn's concern about the novelty of RHGCL. To address this, we added the limitations of existing methods in our revised paper:
>
> *[Section 1] Existing representation clustering methods such as NCL \citep{lin2022improving} and HNECL \citep{wei2025hierarchical} apply contrastive learning between node representations and their clustered prototype representations. This tends to reduce embedding uniformity, which is a property shown to be beneficial for prediction performance in graph contrastive learning \citep{yu2023xsimgcl}*.
>
> Besides, we highlighted the advantage of RHGCL:
>
> *[Section 1] To preserve the uniformity of item node embeddings, we do not include clustered item representations in contrastive learning at this stage.*
>
> *[Section 4] Notably, we do not include the contrastive learning term between item representations and clustered item representations (i.e., $\mathcal{L}_{cl}^{ic}$), as it may compromise embedding uniformity.*
>
> **Reply to Weakness 2 and Question 2.** We sincerely thank Reviewer 4gjn for the reminder of time complexity analysis. To address this, we added Section **Time Complexity Analysis** in Appendix.
>
> Compared with XSimGCL, RHGCL only brings an additional O($2$|$E^{uc}$|$K$$d$) time complexity, which is empirically small given |$E^{uc}$| is small. Here, |$E^{uc}$|, $K$, and $d$ denote the number of edges in the user-clustered item graph, the number of layers, and hidden dimension, respectively.
>
> *Section E. Time Complexity Analysis*
>
> *In this section, we present the time complexity analysis of our proposed RHGCL and contrast it with XSimGCL. To clarify, we first summarize key notations. Recall from Sec. 3 that |$E$| represents the number of edges in the user-item bipartite graph. $K$ is the number of layers in graph convolution operations and $d$ represents the node embedding dimension. We introduce |$E^{uc}$| to denote the number of edges between the user set $U$ and the clustered item set $C$. Hence, we have:*
>
> \begin{equation}
>     |E^{uc}| \leq |E|, |E^{uc}| \leq m\rho\theta.
> \end{equation}
>
> *The first inequality holds since each user-item pair in $E$ is mapped to a user-clustered item pair in $E^{uc}$. The second inequality follows from the fact that the number of edges in $U \cup C$ is upper bounded by the size of $U$ (i.e., $m$) and the size of $C$ (i.e., $\rho\theta$). Further, we use $b$ to represent the batch size and $B$ to denote the number of nodes within a batch.*
>
> |----Model|------------Graph convolution|Recommendation loss|Contrastive loss|
> |XSimGCL|---------------------O($2$ |$E$|$K$$d$)|-----------------------O($2$$bd$)|------------O($bBd$)|
> |---RHGCL|O($2$|$E$|$K$$d$)+O($2$|$E^{uc}$|$K$$d$)|----------------------O($2$$bd$)|------------O($bBd$)|
>
> *Accordingly, we summarize the time complexity of RHGCL and XSimGCL in Table 4. Recall from Sec. 4.3 that we perform graph convolution operations on the user-item graph and the user-clustered item graph separately. This introduces an additional graph convolution time complexity $O(2|E^{uc}|Kd)$, which is theoretically upper bounded by $O(2|E|Kd)$ and is empirically small given the small number of clusters (i.e., $\rho\theta$). Together, RHGCL introduces only a minor increase in time complexity over XSimGCL.*
>
> **Reply to Weakness 3.** We are grateful to Reviewer 4gjn for pointing out the issues with model generalizability, computational complexity, and clustering stability.
>
> *Generalizability.* Our use of t-SNE and radial-angular partitioning have three hyperparameters: perplexity, rho, and theta. Note that any collections of item embeddings can be clustered with the three parameters. Through sensitivity analysis (Figures 3 and 4), we find [10, 200], [1, 8], and [2, 16] contain the optimal parameters of perplexity, rho, theta for the three datasets, respectively. For new datasets, we suggest readers to test hyperparameters starting from these sets.
>
> *Computational complexity.* In Section 4.2, we added the following texts.
>
> *Note that the time complexity of the t-SNE algorithm is O($n^{2}$). This can be optimized to O($nlog(n)$) from the Barnes-Hut SNE algorithm for large graphs.*
>
> *Stability.* We acknowledge that the stability of item node clustering may be an issue for large graphs. This is a common issue for other embedding clustering methods such as K-Means. In practice, we employed the sklearn.manifold.TSNE(*,...) function within the Sklearn Python package with random_state=42.
>
> **Reply to Question 3.** We thank Reviewer 4gjn for the comment regarding the ablation study on the clustering module. Recall that RHGCL extends XSimGCL with the item node clustering component. Hence, the comparison between RHGCL and XSimGCL serves as a direct ablation of the clustering module. Experimental results in Table 2 imply that RHGCL consistently outperforms XSimGCL, demonstrating the positive contribution of the clustering module to model performance.

---

### Official Review · Reviewer_p9ZG · 2025-11-01

**Soundness:** 2
**Presentation:** 2
**Contribution:** 2
**Rating:** 4
**Confidence:** 4

**Summary:**

This paper proposes Representation-driven Hierarchical Graph Contrastive Learning (RHGCL), which enhances user-item recommendation by explicitly modeling hierarchical item structures within a graph contrastive learning framework. Unlike existing GCL methods that overlook multi-level item similarities, RHGCL first pre-trains user and item representations using cross-layer contrastive learning, then clusters items into hierarchical groups via representation compression, and finally fine-tunes embeddings on a two-hierarchy bipartite graph combining both original and clustered interactions. By leveraging both fine-grained and coarse-grained item relationships, RHGCL improves recommendation accuracy, achieving superior performance on benchmark datasets.

**Strengths:**

1. RHGCL learns item hierarchies directly from data-driven representations, enabling multi-resolution modeling without relying on external graphs or attribute labels.

2. The proposed experiments across three benchmark datasets seem to demonstrate the effectiveness of the proposed method

**Weaknesses:**

1. The paper presents item clustering as a key innovation, but similar representation-driven clustering ideas (e.g., neighborhood-enhanced clustering in NCL [1]) have already appeared in recent recommendation literature; the proposed method is not compared with such closely-related works, weakening the novelty claim. Could the authors add comparisons with such methods to further highlight the unique contribution of the proposed clustering module?

2. All contrastive-learning baselines compared are from 2023 or earlier; several stronger 2024/2025 GCL recommenders are ignored, so the performance gap may be smaller than reported.

3. The paper offers no case study or concrete examples to illustrate why the hierarchical module matters, leaving readers without intuitive evidence of its practical value.

[1] "Improving Graph Collaborative Filtering with Neighborhood-enriched Contrastive Learning" Lin et al. WWW 2022

**Questions:**

See Weakness.

---

> ### Author Response · Authors · 2025-12-01
> **Official Reply to Reviewer p9ZG**
>
> **Reply to Weakness 1.** We appreciate Reviewer p9ZG for suggesting a comparison between RHGCL and NCL.
>
> *Experiment comparison.* We carefully read the NCL paper (https://arxiv.org/pdf/2202.06200; WWW 2022) and implemented NCL using code from the SELFRec repository (https://github.com/Coder-Yu/SELFRec/blob/main/model/graph/NCL.py). The hyperparameters are set as their default values in this repository. We summarize the performance below (also in Table 2 in our revised paper). We notice that RHGCL achieves higher prediction accuracy than NCL on the three datasets.
>
> |-------------| ------------------Yelp2018|---------Amazon-Kindle|--------Alibaba-iFashion|
> |----Model|Recall@20|NDCG@20|Recall@20|NDCG@20|Recall@20|NDCG@20|
> |-------NCL|-----0.0674|------0.0553|-----0.1857|-----0.1147|------0.0897|-----0.0417|
> |XSimGCL|-----0.0726|------0.0597|-----0.2061|-----0.1324|------0.1189|-----0.0569|
> |---RHGCL|-----0.0736|------0.0605|-----0.2117|-----0.1354|------0.1194|-----0.0572|
>
> *Analysis.* The above results indicate that NCL underperforms XSimGCL and RHGCL. NCL conducts contrastive learning between user embeddings e_{u} and their clustered prototype embeddings c_{i} (Equation 10 in the NCL paper), and between item embeddings e_{i} and their clustered prototype embeddings c_{j} (Equation 11 in the NCL paper). This approach pushes user and item embeddings closer to their prototype embeddings, reducing the embedding uniformity. Here, embedding uniformity is a property that the learned embeddings are evenly distributed across the space.
>
> Actually, embedding uniformity has been identified as the key factor of performance improvement in the XSimGCL paper (Section 2.3 at https://arxiv.org/pdf/2209.02544; IEEE TKDE 2023). XSimGCL employs random noise injections to directly enhance embedding uniformity, thus achieving superior performance over NCL. RHGCL maintains embedding uniformity through random noise injections. This avoids contrastive learning between node embeddings and clustered prototype embeddings in NCL. This helps explain why RHGCL outperforms NCL.
>
> *Unique contribution.* RHGCL leverages t-SNE while NCL utilizes K-Means. We claim that t-SNE is better suited to user-item recommendation tasks than K-Means, as t-SNE preserves more local structures than K-Means (Reply to Weakness 2 and Question 1 to Reviewer myzr).
>
> **Reply to Weakness 2.** We agree with Reviewer p9ZG that stronger baselines in 2024/2025 should be included. Accordingly, we reviewed the HNECL paper (Knowledge-Based Systems, 2025) and implemented HNECL on the three datasets. HNECL clusters node embeddings into multi-layer hierarchical prototypes and conducts contrastive learning between user and item embeddings on distinct graph convolution layers individually.
>
> For HNECL, we use the same hyperparameters as those in RHGCL and XSimGCL to ensure fair comparison. Furthermore, we set the number of hierarchical prototypes at three layers as (2000, 1000, 100). This configuration follows Table 4 in the HNECL paper. We present the performance comparison below and also update Table 2 in our revised paper.
>
> |-------------| ------------------Yelp2018|---------Amazon-Kindle|--------Alibaba-iFashion|
> |----Model|Recall@20|NDCG@20|Recall@20|NDCG@20|Recall@20|NDCG@20|
> |---HNECL|-----0.0674|------0.0554|-----0.1823|-----0.1096|------0.0976|-----0.0454|
> |XSimGCL|-----0.0726|------0.0597|-----0.2061|-----0.1324|------0.1189|-----0.0569|
> |---RHGCL|-----0.0736|------0.0605|-----0.2117|-----0.1354|------0.1194|-----0.0572|
>
> The results show that HNECL underperforms XSimGCL and RHGCL. As indicated in Equation 14 in the HNECL paper, HNECL performs contrastive learning on node embeddings (e_{u}, e_{i}) and their prototype embeddings (c_{u,l}, c_{i,l}). These operations decrease the embedding uniformity within the representation space, resulting in lower performance compared to XSimGCL and RHGCL.
>
> **Reply to Weakness 3.** We sincerely thank Reviewer p9ZG for questions regarding the mechanism and evidence of hierarchical modules. We are pleased to provide the following clarifications.
>
> *Mechanism.* As shown in Equation 7 of the manuscript, the connecting strength y_{ij} is computed based on the user embeddings e_{U,i}, the item embeddings e_{I,j}, and the clustered item embeddings e_{C,k}. Here, the hierarchical module provides a reference signal (e.g., sports video) that bridges related items (e.g., snooker videos and football videos). This guidance enables RHGCL to achieve higher prediction accuracy compared to XSimGCL.
>
> *Evidence.* As illustrated in Figure 5 (Appendix), the average gap between the connecting strengths of positive and negative user-item pairs increases when the hierarchical module is incorporated into model training. This observation confirms that the hierarchical module enhances the model's ability to distinguish between positive and negative user-item pairs, serving as the evidence for model performance improvement.

---

### Author Response · Authors · 2025-12-02
**Paper Revision After Rebuttal**

We sincerely thank Reviewers p9ZG, 4gjn, and myzr for their efforts to improve the quality of our paper. We have carefully considered and addressed all comments. All revisions are highlighted in red in the revised paper. Thanks again for the valuable input on this paper.

---

### Meta-Review · Area_Chair_L5EM · 2026-01-06

**Summary:**

Despite the RHGCL model's demonstrated effectiveness and clear writing, the paper was rejected due to its limited novelty as a straightforward extension of existing methods, an arbitrary methodological design lacking theoretical grounding, and marginal performance improvements over incomplete baselines without supporting qualitative analysis.

**Reviewer Concerns:**

Mechanism Explanation: The authors attempted to explain how the hierarchical module bridges related items via reference signals, citing appendix figures as evidence.

Weak Novelty: The defense that "t-SNE is better than K-Means" did not fundamentally address the criticism of merely "combining different small pieces." The theoretical justification for the deterministic polar partitioning remains insufficient.

Methodological Arbitrariness: The concerns regarding the introduction of hyperparameters and potential instability from t-SNE and polar partitioning were not effectively countered. Deep comparisons with more advanced hierarchical learning methods are still missing.

**Reviewer Scores:**

none

---

### Decision · Program_Chairs · 2026-01-26

Reject